# Glacial Legacies: Microbial Communities of Antarctic Refugia

**DOI:** 10.3390/biology11101440

**Published:** 2022-10-01

**Authors:** Abigail C. Jackson, Jesse Jorna, John M. Chaston, Byron J. Adams

**Affiliations:** 1Department of Biology, Brigham Young University, Provo, UT 84602, USA; 2Department of Plant and Wildlife Sciences, Brigham Young University, Provo, UT 84602, USA; 3Monte L. Bean Museum, Brigham Young University, Provo, UT 84602, USA

**Keywords:** Antarctica, microbial communities, refugia, metabarcoding, McMurdo Dry Valleys, soil biodiversity

## Abstract

**Simple Summary:**

Microbial communities in Antarctica have only recently been described with the increasing popularity and ease of genome sequencing. Using these methods, we aimed to test hypotheses of refugia using microbial communities in the McMurdo Dry Valleys of Victoria Land, Antarctica. Refugia are habitable areas that remain undisturbed during cycles of glacial expansions throughout Earth’s history. They may contain ancient lineages and unique communities worthy of conservation, as well as provide insight into the biotic history of Antarctica. We found unique microbial community assemblages from putative refugia in the McMurdo Dry Valleys indicating long-lived climax-communities in one of the harshest environments in the world. This finding corroborates the importance of glacial legacies in structuring not just the physical and geochemical environment, but also the soil microbial communities in this landscape.

**Abstract:**

In the cold deserts of the McMurdo Dry Valleys (MDV) the suitability of soil for microbial life is determined by both contemporary processes and legacy effects. Climatic changes and accompanying glacial activity have caused local extinctions and lasting geochemical changes to parts of these soil ecosystems over several million years, while areas of refugia may have escaped these disturbances and existed under relatively stable conditions. This study describes the impact of historical glacial and lacustrine disturbance events on microbial communities across the MDV to investigate how this divergent disturbance history influenced the structuring of microbial communities across this otherwise very stable ecosystem. Soil bacterial communities from 17 sites representing either putative refugia or sites disturbed during the Last Glacial Maximum (LGM) (22-17 kya) were characterized using 16 S metabarcoding. Regardless of geographic distance, several putative refugia sites at elevations above 600 m displayed highly similar microbial communities. At a regional scale, community composition was found to be influenced by elevation and geographic proximity more so than soil geochemical properties. These results suggest that despite the extreme conditions, diverse microbial communities exist in these putative refugia that have presumably remained undisturbed at least through the LGM. We suggest that similarities in microbial communities can be interpreted as evidence for historical climate legacies on an ecosystem-wide scale.

## 1. Introduction

Currently, only 0.34% of Antarctica is ice-free, with the remaining 99.7% covered by ice on average 2 km thick [1]. While the McMurdo Dry Valleys (MDV), located in Southern Victoria Land, East Antarctica, are the largest ice-free region in Antarctica (~4500 km^2^; [2]), the contemporary conditions of the MDV are largely inimical to life. Mean annual temperatures of −20 ℃ and annual precipitation of less than 5 cm water equivalent [3] lead to highly stable, but inhospitable conditions in which soil communities experience little change [4]. 

Soil communities in the MDV are markedly limited to few eukaryotic phyla present only in soils with suitable levels of carbon, moisture, and low salinity [5] and are otherwise dominated by prokaryotic life such as bacteria and archaea [6]. Although microbial taxonomic knowledge is limited for MDV soil communities, molecular-based phylogenetic studies of MDV taxa find relatively high microbial diversity, particularly for those that are desiccation resistant, halotolerant, and psychrotolerant [7,8,9]. Actinobacteria, Bacteroidetes, Proteobacteria, Gemmatimonadetes, Cyanobacteria, and Acidobacteria are thus the most common phyla in the exposed soils of the Dry Valleys [8,10].

The prokaryotic diversity in these remote valleys is largely endemic as there is no evidence of biological movement from other continents or sub-Antarctic Islands currently [11] and through the Cenozoic era [12,13,14,15]. Through cycles of glacial expansion, refugia sites within the MDV that remained undisturbed and habitable, were necessary for endemic organisms’ survival and eventual repopulation of glacially inundated or flooded valleys. The most recent major disturbance event in the MDV is from the Ross Ice Sheet incursion during the Last Glacial Maximum (LGM; 22–17 kya) as it encroached on coastal areas of the MDV and inundated the major valley basins with paleolakes and local glaciers experiencing asynchronous expansions to the global glacial cycles [16,17,18,19,20,21]. These recent landscape disruptions, give each valley a unique geological history [16,19,20]. Large differences in environmental conditions following divergent geological histories has resulted in phylogenetically distinct microbial communities found across the different valley basins in the MDV [10,22].

Within the ice-free soil of the MDV, microbial community structure is constrained by local environmental conditions such as available moisture and soil geochemistry [23]. Soil geochemistry such as salinity, a proxy for soil exposure age [24,25,26], and soil moisture are frequently cited as the strongest controls over community distributions [9,10,27,28,29]. Regionally, variation in community composition is dictated by these independent abiotic environmental conditions. While at local scales, variation in composition is linked to spatial autocorrelation and heterogeneous topography [30]. Connectivity of landscapes [31] and biotic interactions [32,33] are also increasingly recognized as additional drivers of community composition.

Historical legacy effects further influence microbial diversity patterns in the MDV. Specifically, glacial and lacustrine legacies dictate contemporary geochemistry, water availability, and soil organic matter content, thereby creating the heterogeneous landscape for various microbial communities [5,25,31,34]. Experiencing sustained cold desert conditions since the mid-Miocene (23-5 mya), the highest and longest exposed soils in the MDV are very low in microbial biomass due to inhospitable cold-dry-salty conditions [35,36,37,38]. Abundance and biotic complexity decrease with greater exposure age, such that soils exposed through the LGM are generally less suitable for life [39,40,41,42]. Instead, low elevation and limited salt accumulation are the best predictors of soil habitability in these systems [35]. However, even lower elevation soils sustain lower levels of diversity as abundance of carbon depreciates and atmospherically deposited salts accumulate with long exposure ages and minimal periodic wetting [5,26]. Broadly, contemporary microbial distribution in MDV soil is thus a result of regional stochastic colonization processes and the habitability of heterogeneous microenvironments (deterministic processes). These processes, in combination with historical factors, such as the Ross Ice Sheet incursion, produce the observed patchy distributions, variable densities, and co-occurrence patterns observed in the contemporary structure of MDV microbial communities [33].

For most Antarctic regions, including the MDV, refugia sites have been an object of study to determine climax communities, phylogeographic patterns and repopulation rates. The consensus is that biota has persisted in these environments throughout the Pleistocene and earlier cycles of glacial activity [13,43,44,45,46,47,48,49]. Identifying those refugia in the MDV with consistent habitation throughout the LGM requires several considerations. Firstly, soil communities in these locations need to have escaped LGM glacial disturbance. This requires that soils be at high elevations, generally above 1000 m, to have escaped recent glacial disturbance [50], but even some low elevation localities further from the coast have escaped inundation by the expansion of the Ross Ice Sheet and associated paleolakes [51,52]. Secondly, high elevation habitable soils require periodic wetting to temper the negative effects of salt deposition for habitability [53]. Finally, soils need to have been historically colonized and their biota persists through the LGM. Previous work on springtails [46,54,55,56,57], mites [43,58], midges [45], and mosses [48] support evidence of glacial refugia using population-level genetic comparisons between refugia and more recently colonized habitat. Phylogeographic analyses have been used to identify Antarctic refugia, both on Antarctic islands [47] and on the continent itself [44]. Microbial signatures of refugial are expected to reflect these phylogeographic patterns, with putative refugia displaying distinct climax communities and exhibiting phylogenetic similarity when compared to more recently disturbed locations. 

These cold desert ecosystems are particularly sensitive to environmental changes, brief or long-term, due to millions of years of adaptation in severely limited and extreme environments [59]. Climate change has already altered the community composition of microfauna in the MDV, disproportionately impacting species adapted to dry and cold conditions, and ultimately decreasing the overall abundance of soil fauna [60]. Ancient, endemic lineages only persist in patchily distributed habitable spaces [61] and are already at the limits of life in the discontinuous environments of the MDV. Climate change may disproportionally affect these dispersal-limited terrestrial organisms given the lack of ecosystem function overlap in the MDV [52,59,62]. Identifying past and present refugia is an important consideration in understanding the resistance and resilience of the MDV to climate change [63]. This study presents a biological sampling effort of sites in the MDV which may represent putative refugia to determine if there is a microbial community signature comparable to previous patterns observed for invertebrates and mosses in other Antarctic systems, such as higher phylogenetic diversity of microbial taxa in putative refugia sites. We set out to better understand how the structure of microbial communities corresponds to the glacial history of these valleys and identify areas of high biodiversity providing sanctuary to Antarctic life during its climate history. We hypothesize that putative refugia will have distinct microbial communities as a result of climate legacies caused by the LGM.

## 2. Materials and Methods

### 2.1. Sampling Site Selection

We selected sampling sites based on the following criteria: signs of periodic wetting, elevation, published exposure age estimates, and proximity to recorded local glacier terminal moraines or paleolake shorelines. Putative refugia were selected at higher elevations that avoided paleolake inundation and glacial incursion, yet still experienced periodic wetting to temper the negative effects of salt deposition on viable mesofaunal communities [41,53]. Sites disturbed during the LGM either by glacial expansion or paleolake inundation were chosen at lower elevation and with reported soil exposure ages younger than the LGM (less than 26 kya), as well as proximity to recorded local glacier terminal moraines or paleolake shorelines (see Table 1 for site classification). Information from geological surveys dating soil exposure ages was used to discriminate between sites that have remained exposed since before the LGM (>26 kya) as putative refugia, and more recently disturbed sites. However, no precise consensus about which candidate sites represent actual refugia in the MDV exists in the current literature due to an incomplete exposure dating record and limited biological census identifying high elevation habitats. It is likely that refugia existed throughout the MDV as temporarily overlapping habitable soils throughout recurrent glacial cycles of the Pleistocene [13].

Two sample sites were chosen per valley system to incorporate varying geological histories, although not every valley system could be represented by both a putative refugia and disturbed site. Samples were all collected from sites in the MDV during the Antarctic summer, between December and February, in years ranging from 1994 to 2018. For each site, only one soil sample was used for DNA extraction. Figure 1 shows sample locations on a map of the McMurdo Dry Valleys. Site elevation and distance from the coast were estimated using Google Earth.

### 2.2. DNA Extraction and Sequencing

Entire soil top layers to 10 cm depth were removed from one location at each of the sampling sites using the standard sampling procedure for Dry Valley soils described in Freckman and Virginia [73]. All samples were frozen after initial microinvertebrate extractions and were thawed to 10 °C prior to microbial DNA extraction. DNA extraction was performed on each sample using the provided standard protocol with DNeasy PowerSoil Kit (Qiagen, Germantown, MD, USA). For each sample, 0.25 g of homogenized whole soil was used for each extraction, replicated 6-fold per sample. A total of 5 negative controls were also performed on pure PCR-grade water during extraction. One reagent-only control was included in the final sequencing products and one positive control was used to ensure amplification and sequencing worked as planned. Sequencing libraries were prepared from each sample following established protocols [74]. In total, 4 µL of extracted DNA was added to a 96 well plate including sample-specific primers with 17 µL of AccuPrime Pfx Supermix, and 2 µL of each paired set of dual-indexed primers for the V4 region of the 16 S rRNA gene [74]. To prevent evaporation during thermal cycling, a drop of mineral oil was added to each well. Individual 20 µL reactions were placed into a thermocycler at 95 °C for 2 min followed by 30 cycles of 95 °C for 20 s, 55 °C for 15 s, and 72 °C for 5 min then 72 °C for 10 min and held at 4 °C. Amplified DNA was visualized via gel electrophoresis using 4 µL of amplified DNA and 4 µL of loading dye in a 1% agarose gel at 100 volts for 60 min alongside a DNA ladder. PCR amplicons were normalized and pooled to 20 ng/µL per sample using the Charm Biotech Just-a-Plate^TM^ Normalization kit. Amplicons were submitted for sequencing on an Illumina MiSeq (Illumina, San Diego, CA, USA) using paired-end 500 cycle v2 chemistry at the Arizona State University genomics core sequencing center. All post-extraction steps included a reagent-only negative control to remove potential contaminant sequences.

### 2.3. Community Analysis 

All analyses were performed using QIIME 2 (q2) 2021.2 [75]. Raw sequence data from 109 samples plus 5 negative controls and one reagent-only control were first demultiplexed and quality filtered using the q2-demux plugin then denoised with DADA2 denoise-paired function (via q2-dada2) [76]. Sequences were matched with the metadata feature table. Taxonomy was assigned to ASVs using the q2-feature-classifier [77] classify-sklearn naïve Bayes taxonomy classifier against the Silva 138 99% 515 F/860 R region sequences [78]. Sequences were filtered to remove chloroplast and mitochondrial reads with q2’s inbuilt function. Taxonomy bar plots were generated in q2 to visualize diversity at the class-level, excluding samples with fewer than 3 classes represented in total (Figure 2) to remove their outlier effect on beta-diversity metrics. The remaining 86 samples were used to construct a phylogenetic tree with FastTree 2 [79] (via q2-phylogeny) using MAFFT [80] (via q2-alignment) aligned sequences. Based on the observed features, samples were then rarefied (subsampled without replacement) to 2650 sequences per sample, which excluded the negative and reagent-only controls, leaving 81 samples for diversity analysis. Lower Beacon Valley was the only sampling site where all samples had sequencing depths below this filter point and as such all those samples were excluded from further analyses. 

Diversity metrics were all computed using q2-diversity, including alpha-diversity, Pielou’s evenness index [81], Shannon entropy [82], Faith’s Phylogenetic Diversity [83], beta diversity weighted UniFrac [84], unweighted UniFrac [85], Bray–Curtis dissimilarity [86], and Principle Coordinate Analysis (PCoA). The average change in PC1 for each site, overall, was tested for difference from zero using a one-sample t-test with Benjamini-Hochberg false discovery rate (FDR) correction [87]. Significant differences between the mean community composition were calculated using q2’s inbuilt PERMANOVA analysis. 

Lastly, specific analysis of microbiome composition (ANCOM) [88] was performed to assess which taxonomic groups were differentially abundant between samples from different valley basins and between putative refugia and disturbed sites. An ANCOM statistical test generates W-values representing a count of ANOVAs that rejected the null hypothesis that a given ASV is equally abundant per site [88]. Using this method, we tested the significance of each ASV being differentially abundant among our tested groups—those with differing site history and those taken from different valley basins. Differentially abundant taxa were resolved to the best taxonomic resolution against the Silva reference database.

### 2.4. Environmental Analysis

Soil environmental parameters were analyzed to control for the potential effects on microbial community composition. Soil acidity and conductivity are important in driving the abundance of key bacterial groups in Dry Valley ecosystems [89], even on a very fine scale [90], while productivity is largely driven by soil moisture content and freely available organic material [91]. 

Environmental analyses were performed by the BYU Environmental Analytics Lab. In short, gravimetric water content was estimated by drying 25 g of soil at 105 °C and weighing soil dry mass, electroconductivity was measured using a RC-16 C Conductivity Bridge (Beckman Instruments, Brea, CA) and pH was measured using a Thermo Orion Model 410 A+ (Thermo Electron, Waltham, MA, USA). Organic matter content was detected using chromic acid titration [92] and ammonium and nitrate content measured on a Fialyzer 2000 (FIALAb, Seattle, WA, USA). Phosphorus and potassium content was measured using 0.5 M sodium bicarbonate following the Schoenau and Karamonos protocol [93]. 

### 2.5. Statistical Analyses of Alpha Diversity

Qiime2’s built-in Kruskal–Wallis tests were used to compare the diversity between the two groups with different hypothesized glacial histories, and between the different valley basins. The Shannon diversity index was used to assess the differences in alpha diversity between sampled sites. This alpha diversity index was chosen so that abundant taxa and phylogenetic differences between the community members at different sites would be represented in the community analysis. As Shannon diversity had a normal distribution across our sampled locations, the effects of elevation, location, and site geochemistry on microbial alpha diversity were modeled using multiple linear regression models [94] in R [95]. To control for correlative effects of different environmental conditions, a model selection approach was used to select the best explanatory factors for the full dataset as well as the putative refugia sites and the assumed recently disturbed sites separately. The full dataset was considered by including the variations between putative refugia sites and disturbed sites as a random effect, and r-squared values were estimated for the resulting mixed effect models using the piecewiseSEM package in R (https://github.com/jslefche/piecewiseSEM accessed 21 September 2022). As samples were sequenced in up to six replicates, within-location variation was included as a random factor to avoid pseudoreplication. Distance to coast and elevation were strongly correlated in the dataset and so were considered separately to select a best-fitting model. The effects of distance to coast and elevation were further explored in the complete model selection using other environmental factors, which were correlated to these effects in different ways depending on the the site’s proposed glacial history.

### 2.6. Multivariate Analyses

The beta diversity results of the different communities were combined with the effects of our measured environmental variables to plot these interactions in non-metric multidimensional space [96] and using distance-based redundancy analysis [97]. Using the vegan package in R [98] the community composition of each site was plotted in both ordination spaces using the Bray–Curtis distance metric, after which the environmental drivers of this composition were plotted using proportionally scaled arrows. Detrended correspondence analysis [99] was further used to validate the results of these ordination plots.

## 3. Results

### 3.1. Sample Site Classification

Given that our sampling sites were uniformly chosen for habitable soil, that is, signs of periodic wetting and limited salt accumulation, geochemical factors normally used as a proxy for soil exposure age—sulfate, nitrate, and chloride concentrations– were not used to categorize sites in this study [23,24,25,26]. The initial classification of putative refugia or disturbed site status mainly relied on published exposure age estimates (Table 1). Putative refugia sites were identified in all sampled valley basins except the McKay Glacier Basin and Taylor Valley, while disturbed sites were located in all valley basins except the Beacon and Wright Valleys. Putative refugia sites were located at elevations ranging from 646 m (Miers Valley) to 1522 m (Beacon Valley), while disturbed sites were generally located at much lower elevations ranging from 9 m (Taylor Valley) to 873 m (McKay Glacier Basin) above sea level.

### 3.2. Data Retention

After filtering for chloroplast and mitochondrial sequences, samples were filtered to a sampling depth of 2650 sequences per sample based on the alpha rarefaction curve (see Appendix A), retaining 81 samples and 214,650 features (from 1,504,997 total reads). Battleship Promontory, Labyrinth, Levy Cirque, Lower Alatna Valley, Lower Mt Suess, and Wall Valley each retained *n* = 5. Brownworth, Higher Miers Valley and the Dais each retained *n* = 3. Miers Lake, Hawkings Cirque and Lower Taylor Valley retained *n* = 4. All other samples retained *n* = 6. Lower Beacon Valley was filtered out due to low sequencing depth. Retained samples thus represented 17 total sites containing 8 paired high and low elevations, 6 of which are hypothesized to have escaped recent glaciation and the others having been recently disturbed in the LGM. Taxonomic composition across these samples at the class level are presented in Figure 2. A total of 1390 unique features were retained after rarefaction, and 94 different classes were identified across our samples, with the top 3 classes being Actinobacteria, Blastocatellia and Thermoleophilia, together making up around 40% of total reads (Figure 2). Actinobacteriota, Acidobacteriota, Chloroflexi, and Alphaproteobacteria were the four most prevalent phyla of the 33 identified, together making up at least 40%, but frequently over 60% of the reads in each sample after filtering. Crenarchaeota and Halobacterota were the only clades of Archaea recovered from these samples.

### 3.3. Alpha Diversity

Shannon diversity was highest in the McKay Glacier Basin and Garwood Valley, followed by Miers Valley, but significant differences in diversity were found within some of these valleys (Figure 3). For example, significantly higher diversity was found in the disturbed site within Miers Valley than in the higher elevation putative refugium in the same valley basin (*p* < 0.05, Kruskal–Wallis pairwise comparison). The full table with Shannon diversity comparisons between each site is reported in Appendix A. Overall, a higher alpha diversity was found in recently disturbed sites over putative refugia sites and this distinction was found to be significant between the two groups (*p* = 0.003, Kruskal–Wallis pairwise comparison). These results were consistent between the Faith PD metric and Shannon diversity, but not when calculating Pielou’s evenness. Faith-phylogenetic distance and Pielou evenness for putative refugia and recently disturbed samples are reported in the Appendix A.

### 3.4. Beta Diversity

Community composition (measured in the Unweighted UniFrac distance between samples) was significantly different between all the sampled valley basins, and each sampled location was found to be significantly different from all other sampled sites in pairwise comparisons, with three sites as exceptions (Dais, Brownworth and Higher Miers Valley, *p* < 0.05 threshold, PERMANOVA). These three comparisons also had the lowest sample size out of all pairwise comparisons. Comparisons with other measures of community dissimilarity were largely congruent except for two exceptions when using the Weighted UniFrac distance to compare different valleys (between Alatna Valley and Upper Wright Valley as well as between Upper Wright Valley and Victoria Valley) (*p* < 0.05 threshold, PERMANOVA test, see Appendix A for reported *p*-values). However, overall, the community composition of putative refugia sites was found to be significantly different from the communities of disturbed sites using all three metrics (*p* = 0.001 (all metrics), PERMANOVA test). Figure 4 shows the community composition of our samples in ordination space, measured in the Unweighted UniFrac, Weighted UniFrac and Bray–Curtis distance metrics.

### 3.5. Effects of Environmental Conditions

#### 3.5.1. Distance to Coast and Elevation

Figure 5 shows the relationship between the distance of a sampling site to the nearest coastline and its elevation above sea level to the Shannon diversity index of the microbial community. The best models selected were discordant between the full dataset and refugia sites and disturbed sites separately, with increased distance to the coastline selected as the best significant predictive factor (*p* < 0.05, linear regression) only in the full dataset, and is associated with a loss of diversity (between −0.012 and −0.026 decrease in Shannon index per kilometer distance from shore). Both the refugia and disturbed site datasets separately show a moderate negative trend in diversity at greater distances from the coast (see Appendix A for full *p*-value and intercept report).

#### 3.5.2. Soil Geochemistry

None of the investigated environmental variables were significantly different between the grouped putative refugia sites and the recently disturbed sites (see Appendix A, *p* > 0.05, ANOVA)—full geochemical data are reported in Appendix A. The correlation matrices for all investigated environmental variables including elevation and distance to coast (see Appendix A) showed strong correlations between several of the key environmental drivers. A model selection approach was used to determine the best predictors of microbial diversity in our datasets. Soil organic matter content was retained as the only factor significant in the full dataset (*p* = 0.0135, “Shatterthwaite’s *t*-test”) while it was not significantly predictive of alpha diversity in the dataset of recently disturbed sites (*p* > 0.05, linear regression) or potential refugia. Figure 6 shows the organic matter content across the two datasets and its relation to alpha diversity. Putative refugial sites and disturbed sites considered separately did not fit in any single linear model, as many of the correlated variables such as distance to coast, elevation and organic matter content had a similar weight in the model selection. Model averaging of all best predictive models in the potential refugia dataset provided a model without any significant effects on alpha diversity that included distance to coast, elevation, organic matter content, gravimetric water content and potassium content. The best predictive single-variable model included only distance to coast, but its effect on diversity was not significantly different from zero (*p* > 0.05, linear regression). In the dataset of disturbed sites, potassium was the only factor retained in the best predictive model but had no significant effect on microbial diversity. All resulting models only explained a small proportion of the dataset’s total variability (adjusted R-squared: 0.19 in the dataset of putative refugia, 0.01 in the dataset of recently disturbed sites, and 0. 33 in the full dataset, see Appendix A for full results). Salt and nitrate concentrations were not significantly correlated with biodiversity when considering the entire dataset or either group of sites separately. Putative refugia sites also did not have any higher average salt concentrations or nitrate content, regardless of their longer exposure age and higher elevation (see Figure 7). Any non-significant environmental predictors across sites are reported in Appendix A.

### 3.6. Drivers of Community Structure

#### 3.6.1. Ordination Plots

Figure 8 shows the result of the non-metric multidimensional scaling (NMDS) plot. Community distance metrics are plotted using the Bray–Curtis metric, reflecting the non-phylogenetically weighted distribution of sampled communities in ordination space. Similar to the alpha diversity models, organic matter plays a large role in driving community structure. Elevation and distance from the coastline have a similar and strong effect on community composition, aligning with the ordination of several of our hypothesized refugia sites such as those in Alatna Valley, Upper Wright Valley and Victoria Valley. Nitrogen content and conductivity had a strong and equal effect on community composition, reflected mostly in the high salt content soils of the Brownworth samples taken from Lower Wright Valley. Geographically, we saw a reasonable effect of longitudinal distance which was inversely correlated compared to the distance from coastline. Latitudinal differences across these valleys did not play a strong role in shaping community structure. Distance-based RDA ordination mapping of environmental drivers corroborated these results and showed highly similar patterns in community structure (see Appendix A). A detrended correspondence analysis showed largely congruent clustering of communities largely driven by a higher degree of heterogeneity of the potential refugia sites (Appendix A).

#### 3.6.2. Geography

Figure 9A plots the first principal coordinate of the Unweighted UniFrac diversity metric against each site’s latitude. Unweighted UniFrac distances were selected as the most informative beta diversity measurement for this figure as it considers the phylogenetic dissimilarity between different communities, thus including a signal of evolutionary history or dispersal limitation between the valleys and over the glaciation history of the sites. This further illustrates those sites, especially those classified as putative refugia, showing remarkably similar community composition over a large latitudinal range. Conversely, sites at higher elevations also had highly similar community composition whereas the lower elevation disturbed sites display a larger variation on the first principal coordinate axis (Figure 9B).

#### 3.6.3. Sample Composition

The analysis of the composition of microbes (ANCOM) comparing putative refugia sites to recently disturbed sites found eleven distinct ASVs that significantly differed in abundance between those two groups. Eight of these ASVs belonging to the Gemmatimonadaceae, Chitinophagaceae, Blastocatellaceae, Gaiellaceae, and Armatimonadales families, and the phylum Chloroflexi are found in significantly higher percentile abundances for putative refugia sites (W = 4717, 4649, 4699, 4452, 4401, 4498, 4644, and 4411, respectively). Two ASVs, both from the order Pyrinomonadales, are associated with recently disturbed sites (W = 4722 and 4532). One archaeal ASV belonging to genus *Candidatus* Nitrocosmicus (W = 4784) also differs between putative refugia and recently disturbed sites. Inter-valley comparisons resolved 517 ASVs (11%) as being significantly associated with a specific valley system. The full table displaying individual taxa ANCOM scores is reported in the Appendix A.

## 4. Discussion

The MDV provide a unique ecosystem to study phylogeographic patterns left by geological cycles and thousands of years of evolution in an environment that has undergone remarkably little change [100]. At the same time, it is well documented that populations of Antarctic biota have been strongly influenced by events in their geological and climate history and carry evolutionary traces of extinction during glacial advance or persistence in non-glaciated refugia [12,13,101]. In this study the microbial composition of several high elevation putative refugia and more recently disturbed low elevation sites were characterized to determine signatures of divergent glacial legacies on microbial communities. 

Geological work has determined that some high elevation sites in continental Antarctica have remained exposed through periods of glacial expansion while low elevation sites were inundated by local glaciers or paleolakes, assumptions supported by Antarctic glacial models, isotope dating, and microarthropod and moss population genetic evidence [43,46,48]. Here, we have identified a biological line of evidence for the presence of such refugia in the Dry Valleys by characterizing the communities of microbial life at putative refugia sites.

### 4.1. Alpha Diversity

While microbial community diversity was initially hypothesized to be higher in refugia, comparable to high population genetic diversity in refugia found among other terrestrial organisms [15,44], our findings present the opposite result. Shannon diversity indices were significantly lower in putative refugia sites than in more recently disturbed sites in this study, while differences between the valley basins were more pronounced than between disturbed sites and putative refugia. None of the individual comparisons showed higher diversity in the putative refugia site compared to a recently disturbed site in the same valley basin, except in Miers and Victoria Valley.

The results of our model selection to determine the effects of environmental factors on alpha diversity across the sample sites showed no clear predictive effects of site biogeochemistry or other environmental factors, such as elevation or distance to the coastline. While in all of the datasets, distance from the coast was found to be a better predictor of decreased biodiversity than elevation, these were non-significant drivers of alpha biodiversity. These patterns were most pronounced in the full dataset, but are likely caused by the location of putative refugia sites, which are all located further from the coastline and at higher overall elevation than the disturbed sites. The geochemical soil properties most predictive of biodiversity were not congruent between putative refugia and disturbed sites, although organic matter content patterns were influenced by zero-inflation, caused by the very low contents found in these Antarctic soils and the limits of the detection test. The nitrate and salt concentrations of soils sampled in this study did not present a clear distinction between longer exposed, putative refugia sites and disturbed sites. This may indicate that many of the high elevation putative refugia investigated in this study are in fact experiencing periodic wetting at a similar frequency to the lower elevation disturbed sites. However, an overall pattern of decreasing alpha diversity with increasing nitrate and salt contents was observed in this study, consistent with the findings of Dragone et al., [35].

Contrary to the widely accepted influence of soil moisture, water limitation did not seem to be a predictive factor in the diversity between the sites sampled in this study, while across the MDV ecosystem this is often suggested as the main constraint on microbial diversity [28,102]. Instead, organic matter content was the best predictor when modeling the lower elevation disturbed sites which are likely to have fewer constraints in water input due to periodic wetting by snow and glacier runoff, ephemeral streams [6], and a closer water table, whereas water availability may be more strongly limited in higher elevation sites. These results are largely congruent with literature showing that while environmental factors are key drivers of soil habitability, their effects are difficult to generalize over large spatial ranges [30,90].

### 4.2. Similarities in Community Composition

Displaying community structure in ordination space did not result in a clear clustering of all disturbed and putative refugia sites across the whole dataset. Instead, several different clustering patterns can be identified which cast doubt on the classification of some of the selected disturbed and putative refugia sites. Strikingly, the factors displaying the strongest effect on community composition are those directly associated with refugia identification: elevation, distance from coast, and the accumulation of nitrate and salt compounds. Patterns of community clustering were consistent regardless of beta diversity metric and ordination type, indicating that these results are robust and microbial community signature is different both in terms of abundance and composition.

### 4.3. Within Valley Comparisons

Across all beta diversity indices, Garwood Valley samples clustered strongly together and were highly similar, although the distances between these communities were reported as significant. The two sites within Garwood Valley also exhibited equally high levels of biodiversity and very similar environmental conditions. Evidence exists for the inundation of our lower elevation site in Garwood Valley [20]. While it is unclear whether higher Garwood Valley escaped inundation or another large disturbance in the relatively recent past, the community signature we describe is highly similar to the lower elevation site in the same valley basin and resembles other disturbed sites from this study.

Conversely, samples from the McKay Glacier region showed large community composition differences although the two sites are at very similar elevations, potentially driven by high organic matter content in only one of the sites. Other environmental variables such as phosphorus content and water availability also differed strongly between the two sites. Despite these differences, both sites exhibited relatively high biodiversity making it hard to draw specific conclusions about the glacial history of these sites, even though relatively recent glaciation has been proposed for this area [70].

In Miers Valley, Taylor Valley, Lower Wright Valley, and Alatna Valley the ordination plots showed high differentiation between the two sampled sites within each valley. These differences were less pronounced in the Unweighted UniFrac distance metric than in the other two beta diversity metrics. This can be partially explained by the large distance between the paired sample locations, such as in the case of Victoria Valley and Lower Wright Valley. However, this is not the case for other within-valley comparisons. Differentiation in Lower Wright Valley samples could be further explained due to large differences in elevation, distance from the coastline, and soil geochemistry, particularly the high concentrations of nitrates and salts found in the disturbed site. The two sites in Taylor Valley similarly have striking differences in local geochemistry and elevation. However, paired samples in Miers Valley exhibit highly similar environmental conditions, with the only difference being the hypothesized inundation of the lower elevation site through glacial lake formation during the LGM (20-10 kya) [20].

### 4.4. Putative Refugia Sites

In each of the plots, a tentative cluster can be found of at least five core putative refugia sites: Battleship Promontory in Alatna Valley, Beacon Valley, both Upper Wright Valley sites and Wall Valley in Victoria Valley. These sites are spread across a large latitudinal range and have well established exposure age estimates that indicate all these sites have remained undisturbed for at least several million years. In each beta diversity metric, the much lower elevation site in Lower Victoria Valley fell into this cluster, supporting the geological evidence that this site has likely been exposed for 120-300 kya, which is prior to the LGM but suggests a more recent disturbance than the core putative refugia sites [19,71,72]. Given that this relatively recently disturbed site clustered with putative refugia sites for which much older exposure age estimates exist, perhaps any exposure age beyond the LGM is a benchmark for a climax community. As evidence, both sites in Victoria Valley clustered closer to the putative refugia sites despite marked dissimilarity resulting from differences in elevation and soil geochemistry.

Clustering patterns for the lower Alatna Valley site were similar to putative refugia, although here, no specific estimates of soil exposure ages have been published. As such, this is a tentative biological estimate that exposure ages in lower Alatna Valley may be similar to those found at Battleship Promontory. The lower elevation site in Alatna Valley was also found to have some of the highest salt and nitrate concentrations in this study, further evidence that these soils have been undisturbed for a considerable amount of time. 

The community composition in high elevation sites from lower Wright and Miers Valleys clustered together with the five core putative refugia sites in the unweighted UniFrac principal coordinate plot, but had varying results using other beta diversity metrics. Regarding putative refugia in Miers Valley, the ambiguity of this classification can be attributed partially to it being located at the lowest elevation out of all the putative refugia sites and at the greatest distance from the rest of the high elevation sites in this study. The Dais site in lower Wright Valley matches the geochemical and elevational conditions of the other putative refugia sites, and geological evidence show this site has been relatively undisturbed for around 4 million years [69]. However, biodiversity was the lowest out of any investigated site in this study, potentially explaining the poor clustering with other putative refugia using both abundance-weighted diversity metrics.

Surprisingly, both the low and high elevation sites from Taylor Valley showed tentative clustering with the larger putative refugia site cluster, especially in the two reported UniFrac ordination plots. While Taylor Valley has been the focus of much study regarding the microbial life in the MDV, specific glacial histories are not known for some of the higher plateaus around its lake basins. Part of the within valley differences are shown to be explained by the large variation in phosphorus availability and pH between the two sites, yet the similarity of both these sites to several of the putative refugia for which most geological evidence exists is puzzling. The lower elevation site in Taylor Valley was almost certainly recently inundated by glacial lakes [16], while the recent history of interactions between Mt. Falconer and the Commonwealth Glacier is less clear.

### 4.5. Similarities across the MDV

Geographic distance plays a strong role in community composition. Beta diversity comparisons among paired sites found that each valley system was significantly distinct from other paired valley sites. Likewise, ANCOMs found significant differences in the dominant taxa among the paired sites of each valley system. While katabatic winds can be a source of dispersal in this polar ecosystem, successful migration is still limited by distance, geographic barriers and local conditions [11,103]. Interestingly, several of the hypothesized refugia sites in this study displayed highly similar microbial communities while being geographically separated. The area from Beacon Valley to Alatna Valley covers the majority of the MDV latitudinal variation, a horizontal distance of more than 100 km with major geographic barriers. Yet, all of the putative refugia sites identified across this region show remarkably similar communities in the context of their geographic distance.

The high degree of similarity between the putative refugia sites in Alatna Valley, Beacon Valley, Miers Valley, Victoria Valley, and both Upper Wright Valley pairs appears not to be structured by commonalities in environmental variables. Unsurprisingly, high elevation and large distance from the coastline are strong predictors of microbial community structure, but even sites with disparate elevations such as the putative refugia in Miers Valley and the lower Victoria Valley site display a strong similarity to the other putative refugia sites at higher elevations.

The similarity in community composition of the paired putative refugia and disturbed site in Garwood Valley presents an argument that this site’s community is not structured by the same legacy effects as the main cluster of putative refugia sites. The other main outlier in this comparison is formed by the low elevation site in Taylor Valley, which exhibits remarkable similarity to the cluster of refugia sites, and the identified putative refugia in Miers Valley which are not as similar to the other putative refugia sites as those within the observed cluster. It also provides increased evidence for the consideration of the lower Victoria Valley site as a putative refugium due to its high degree of similarity to other refugia over a large geographic range.

### 4.6. Taxonomic Differences between Putative Refugia and Disturbed Sites

While metabarcoding approaches have made the description of taxa in extreme environments more accessible [104], functional information for many of these species is dependent on notoriously difficult culture-based approaches [105] and is thus frequently absent from the literature. What literature exists for the dominant families associated with putative refugia revealed by ANCOM are known to have adaptations for low soil moisture, broad pH range, and high salt conditions. For example, bacteria of phylum Gemmatimonadetes are adapted to drier soils [106], the family Blastocatellacae is slightly acidophilic to neutrophilic mesophiles adapted to a broad range of temperatures and pH levels [107], and the phylum Chloroflexi is known to be photoautotrophic, a beneficial attribute given the low carbon input in the MDV [108]. Of the taxa found in higher abundances in recently disturbed sites, the ammonia-oxidizing archaea *Candidatus* Nitrocosmicus could indicate more productivity in these lower valley soils associated with the increased presence of functional nitrification taxa [109]. Eleven percent of all identified ASVs were associated with a specific valley system, indicating a high endemicity of each valley community created by geographic barriers and limited dispersal.

## 5. Conclusions

This study has characterized putative high elevation refugia, capable of supporting microbial diversity comparable to model habitats from low elevation valley sites. As the Antarctic continent is extremely harsh and inhospitable, these islands of suitable soil habitat at high elevations could be considered important sites for management and conservation. If these refugia have insulated microbes from climate changes in the past, they can potentially serve the same purpose in the face of climate shifts. Likewise, the fact that refugia are isolated may provide resistance against potential invasive species that could threaten more vulnerable soils—those at lower elevations and closer to the coast. Refugia may also serve to preserve highly adapted ancient genotypes whereas lower valley basin populations could undergo strong selection as a result of anthropogenic influence. The differentially abundant taxa found by an analysis of microbial composition further supports the hypothesis that these refugia harbor genetically distinct bacterial lineages. 

We show that sites with different glaciation histories have a distinct microbial community composition underscoring the importance of glacial legacies in this system. Putative refugia have distinctive microbial communities when compared to recently disturbed regions. Given the slightly different signal for distinct community compositions between the two glaciation histories using the weighted or unweighted beta diversity measurements, differential ASV abundance may play a stronger role in distinguishing the community composition of putative refugia sites and recently disturbed sites. Community composition was most strongly influenced by elevation and location of the sampled area, but not soil geochemistry. While there may not be one distinct signature of microbial composition in Dry Valley refugia, we show that microbial community structure reflects the glacial history of the McMurdo Dry Valleys. 

## Figures and Tables

**Figure 1 biology-11-01440-f001:**
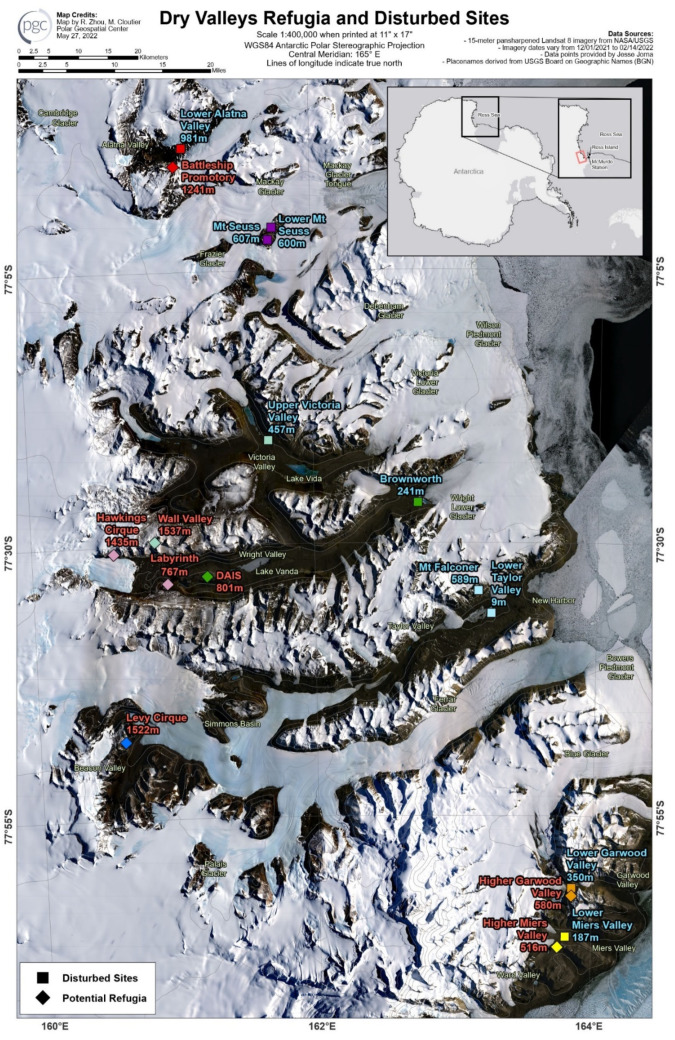
Map of the McMurdo Dry Valleys (MDV) and sampling sites. All 17 samples are displayed at their approximate location of origin. Sites that are classified as disturbed sites are displayed as a square symbol and are marked with blue text, and putative refugia are displayed as diamonds and accompanying red text. Colors of the site markers indicate which valley basin samples came from.

**Figure 2 biology-11-01440-f002:**
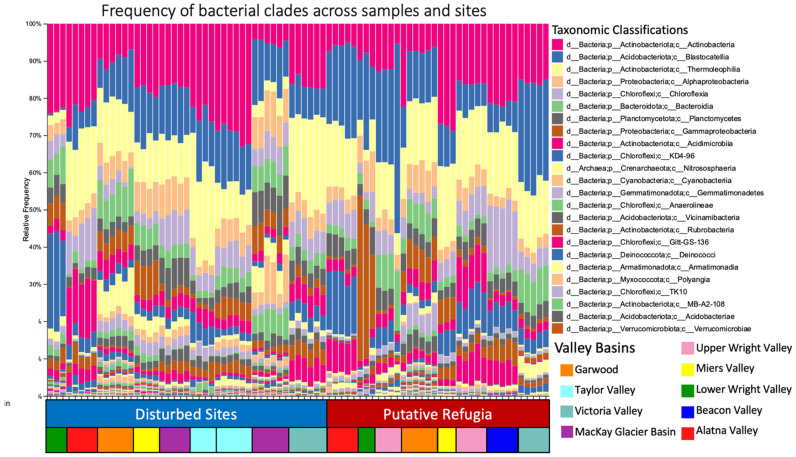
Frequency of bacterial and archaeal clades across samples. Relative frequency of ASVs in each sample is displayed at the class level; the vertical axis shows the proportion of total ASVs in each sample for each class. The horizontal axis is coloured to show if a sample originated from a site characterized as a disturbed site or putative refugia. Colors represent the valley basin of sample origin.

**Figure 3 biology-11-01440-f003:**
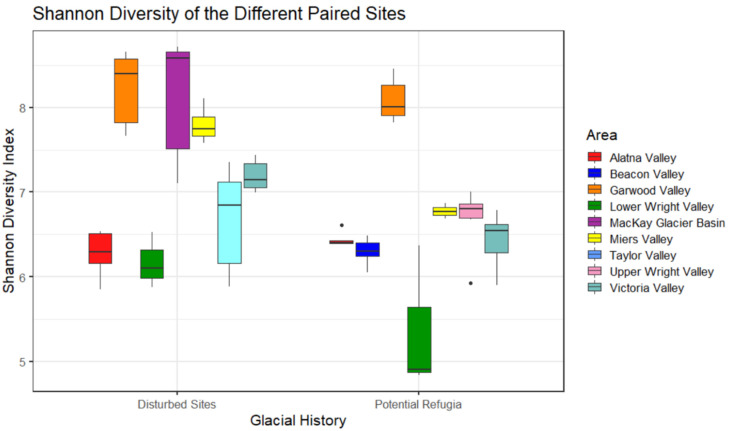
Shannon diversity of the different paired sites. Boxes are made up of all replicates from a single site and show the average diversity. Colors indicate valley basin of site location.

**Figure 4 biology-11-01440-f004:**
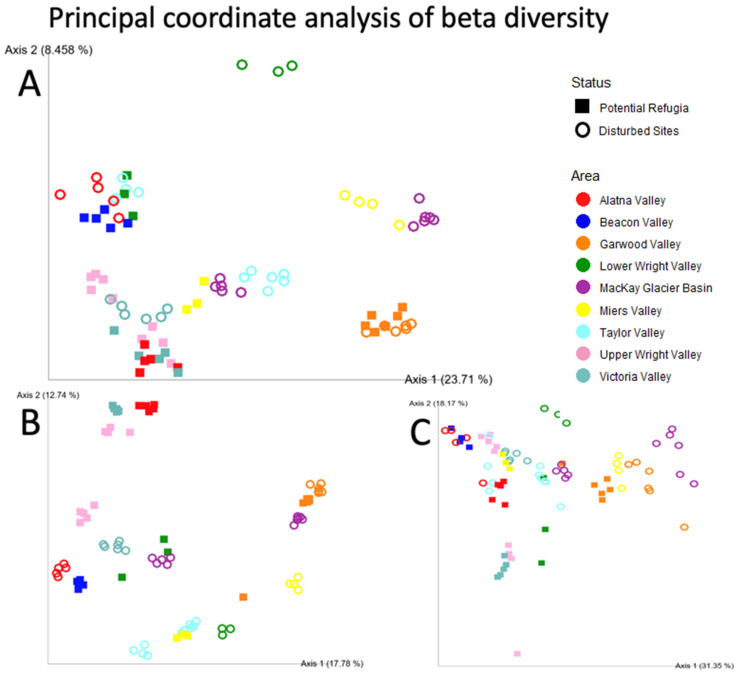
Principal coordinate analysis of beta diversity. Principal coordinate analysis of the beta diversity of each individual sample replicate for (**A**) Unweighted UniFrac distance, (**B**) Bray–Curtis distance and (**C**) Weighted UniFrac distance. Axes labels indicate the weight of the principal co-ordinate. Samples taken from putative refugia sites are shown as squares, disturbed sites are shown as open circles. Colors indicate the valley basin samples originate from.

**Figure 5 biology-11-01440-f005:**
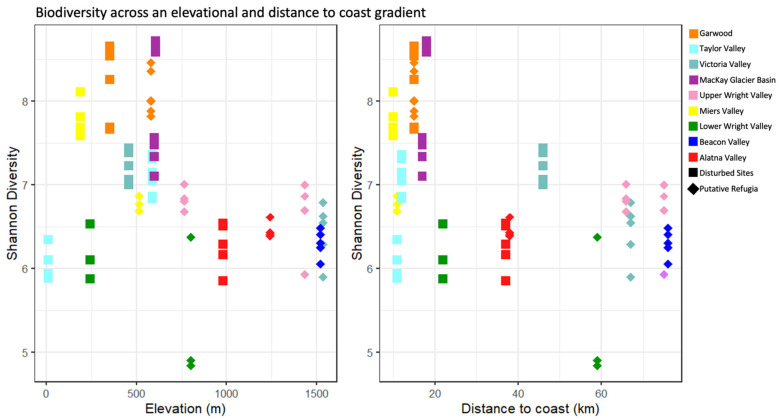
Biodiversity across an elevational and distance to coast gradient. Each individual sample replicates’ Shannon diversity is displayed in relation to the location’s elevation (**left**) and distance to coast (**right**). Diamond shaped points indicate putative refugia and squares indicate disturbed sites, while colors indicate the valley basin of sample origin.

**Figure 6 biology-11-01440-f006:**
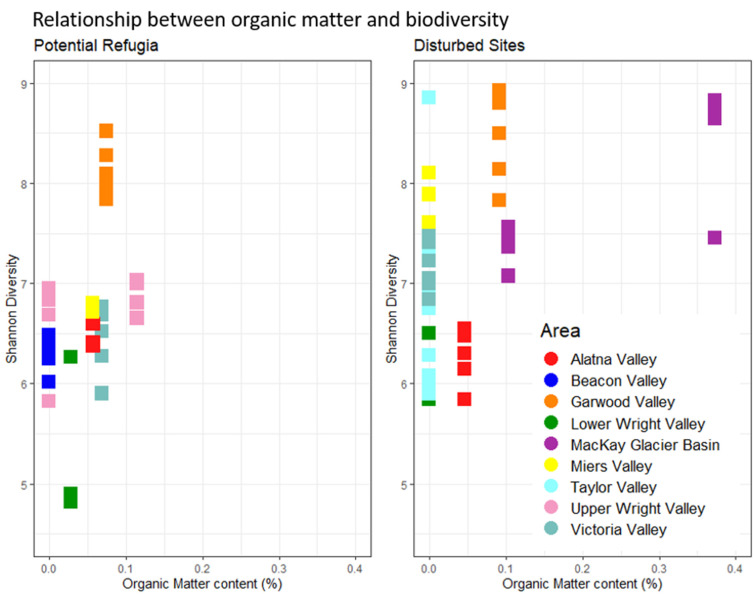
Relationship between organic matter and biodiversity. Individual sample replicates’ Shannon diversity is displayed in relation to the location’s organic matter content. Replicates from the same site share the same organic matter content. Putative refugia sites and disturbed sites are displayed separately. Colors indicate valley basin of sample origin.

**Figure 7 biology-11-01440-f007:**
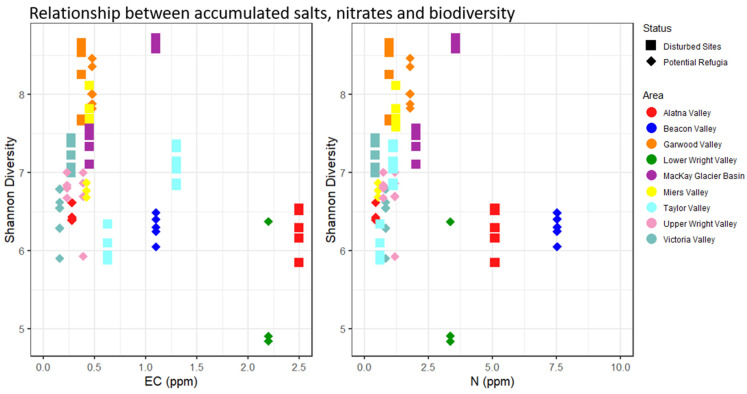
Relationship between accumulated salts (electrical conductivity), nitrate and biodiversity. Individual sample replicates Shannon diversity is displayed in relation to the location’s salt (**left**) and nitrate (**right**) content. Replicates from the same site share the same concentration of salt and nitrate. Diamond-shaped points indicate putative refugia, squares indicate disturbed sites. Colors indicate valley basin of sample origin.

**Figure 8 biology-11-01440-f008:**
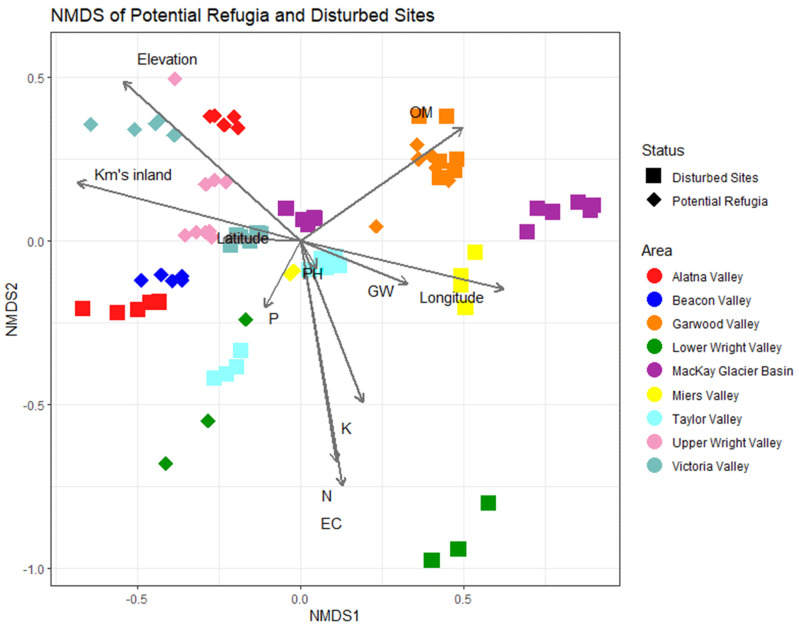
Non-metric multidimensional scaling (NMDS) of putative refugia and disturbed sites. Non-metric multidimensional scaling plot of Bray–Curtis distances between sites. Each point represents a single sample replicate, with diamonds indicating putative refugia sites and squares indicating disturbed sites. Gray lines indicate the correlation between community composition and environmental variables, where longer lines indicate a stronger correlation. Point colors indicate the valley basin the sample originated from.

**Figure 9 biology-11-01440-f009:**
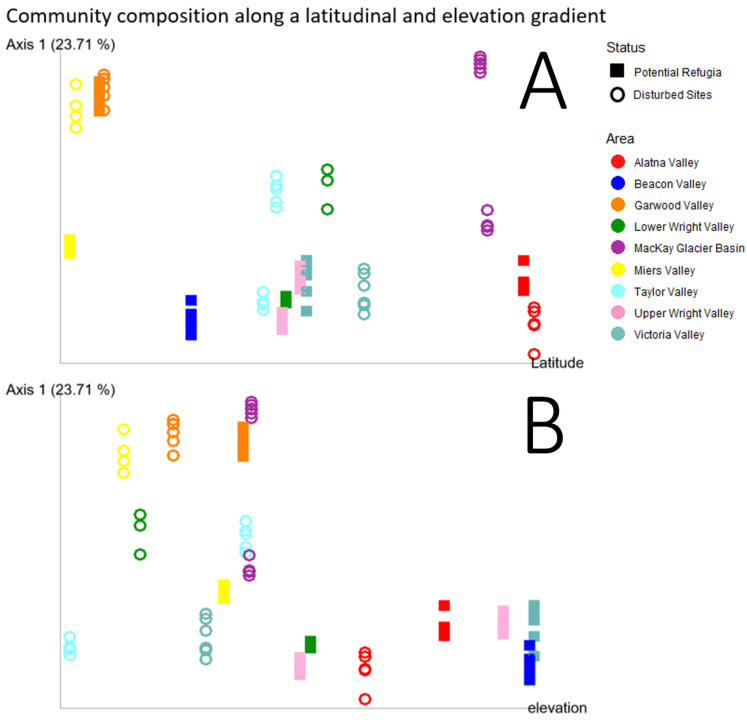
Community composition along a latitudinal and elevational gradient. Principal coordinate analysis of the Unweighted UniFrac diversity metric of each individual sample replicate with the principal coordinate with the highest weight on the vertical axis and latitude (**A**) or elevation (**B**) plotted on the horizontal axis. Samples taken from putative refugia sites are shown as open circles, disturbed sites are shown as squares. Colors indicate the valley basin samples originate from.

**Table 1 biology-11-01440-t001:** Site locations and classification. Location and classification for each site sampled for this study. Note that DNA yields from Lower Beacon Valley were too low to be included in the study. Geological history is based primarily on exposure age estimates from the literature. Citations refer to the primary sources with referenced exposure age estimates for that site.

Valley System	Name	Date Collected	Geologic History	GPS Waypoints		Elevation (m)	Estimated Exposure Age	Citation	Classification
				Lat	Long				
Beacon Valley	Levy Cirque	2018–2019	Putative Refugia	77°47′50.04″ S	160°36′55.72″ E	1522	2.3 mya	[64]	Putative Refugia
	Lower Beacon Valley	2018–2019	Putative Refugia	77°48′39.40″ S	160°41′19.33″ E	1017			
Alatna Valley	Battleship Promontory	2003–2004	Putative Refugia	76°55′20.46″ S	161°4′53.16″ E	1241	5 mya	[65,66,67,68]	Putative Refugia
	Lower Alatna Valley	1995–1996		76°53′40.74″ S	161°8′20.94″ E	981			Putative Refugia
Upper Wright Valley	Hawkings Cirque	2010–2011	Putative Refugia	77°30′36.69″ S	160°34′42.95″ E	1204		[69]	Putative Refugia
	Labyrinth	2005–2006	Putative Refugia	77°33′27.41″ S	160°57′16.49″ E	767	3 mya		Putative Refugia
Lower Wright Valley	Dais	2005–2006	Putative Refugia	77°32′51.96″ S	161°14′11.10″ E	801	4 mya	[18,69]	Putative refugia
	Brownworth	2005–2006	Disturbed	77°26′19.08″ S	162°43′43.69″ E	241	26-5 kya		Disturbed
Mount Suess	Mount Seuss	2010–2013	Disturbed	77°2′11.86″ S	161°42′36.43″ E	706	25-5 kya	[70]	Disturbed
	Lower Mount Suess	2008–2009	Disturbed	77°1′4.11″ S	161°44′18.85″ E	517			Disturbed
Taylor Valley	Mount Falconer	2001–2003	Disturbed	77°34′23.42″ S	163°9′30.95″ E	731		[16]	Disturbed
	Taylor Valley (Lake Fryxell)	2001–2003	Disturbed	77°36′27.70″ S	163°15′2.46″ E	9	21-12 kya		Disturbed
Miers Valley	Higher Miers Valley	2011–2012	Putative Refugia	78°7′0.02″ S	163°45′2.85″ E	516		[17,20]	Putative Refugia
	Lake Miers	2011–2012	Putative Refugia	78°6′2.15″ S	163°48′32.83″ E	167	26-5 kya		Disturbed
Garwood Valley	Higher Garwood Valley	2011–2012	Putative Refugia	78°2′18.96″ S	163°51′2.17″ E	581		[20,21]	Disturbed
	Lower Garwood Valley	2011–2012	Disturbed	78°1′35.32″ S	163°51′4.35″ E	351	26-5 kya		Disturbed
Victoria Valley	Wall Valley	2010–2012	Putative Refugia	77°29′37.02″ S	160°52′15.71″ E	1617		[19,71,72]	Putative Refugia
	Upper Victoria Valley	1993–1994	Disturbed	77°20′31.64″ S	161°41′13.28″ E	457	120–300 ka		Putative Refugia

## Data Availability

Data for this study is available at doi:10.6073/pasta/3380695f50e8f3527c12aba416e86b1f [110].

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
