# Peer review of "Glacial Legacies: Microbial Communities of Antarctic Refugia"

_biology, 2022, doi:10.3390/biology11101440_

Round 1

Reviewer 1 Report

The paper by Jackson et al. describes microbial community structure and a chemical soil background supported by glaciation data in McMurdo Dry

Valley system. The paper is well written. Remarkable is the number of samples investigated which gives a superb overview of soil microbial communities

in these sites. The paper has very few shortcomings. Most notably it lacks a clear aim and a working hypothesis – what were the authors expecting from

this investigation? Overall a very good paper, relevant to the field of microbial ecology.

Some minor issues:

L13 – find? Found?

L52 – exposed soils of the (???)

L53 – remove? Remote?

L496 – are or were influenced?

L601 – katabatic winds

L626 and further – is the title appropriate for this paragraph? Why is the paragraph in bold font? Some taxa are not in written in cursive and with a

capital letter. Please revise this paragraph thoroughly!

L656 – warming? Rather influence in general

L657 – this hypothesis was not mentioned earlier. Is this the working hypothesis for this investigation?

Author Response

Reviewer 2

The paper by Jackson et al. describes microbial community structure and a chemical soil background supported by glaciation data in McMurdo DryValley system. The paper is well written. Remarkable is the number of samples investigated which gives a superb overview of soil microbial communities in these sites. The paper has very few shortcomings. Most notably it lacks a clear aim and a working hypothesis – what were the authors expecting from this investigation? Overall a very good paper, relevant to the field of microbial ecology.

Some minor issues:

L13 – find? Found? corrected

L52 – exposed soils of the (???) corrected

L53 – remove? Remote? corrected

L496 – are or were influenced? corrected

L601 – katabatic winds corrected

L626 and further – is the title appropriate for this paragraph? Why is the paragraph in bold font? Some taxa are not in written in cursive and with a capital letter. Please revise this paragraph thoroughly!

Thank you for pointing out these errors - we have adjusted the text following including new data in the analyses and addressed all these problems.

L656 – warming? Rather influence in general

Corrected

L657 – this hypothesis was not mentioned earlier. Is this the working hypothesis for this investigation?

Thank you for the suggestion. We have tried to address our hypotheses by adding a line in the abstract, and in L155-156 of the introduction. We have changed the wording of this particular phrase in L692 to be more appropriate for the analyses actually performed.

Reviewer 2 Report

The manuscript by Jackson et. al entitled “Glacial legacies: Microbial communities of Antarctic refugia” describes the impact of historical glacial and lacustrine disturbance events on microbial communities in the McMurdo dry valleys (MDV). They have reported the effect of historical glacial and lacustrine disturbance events on microbial communities. They also identified similar microbial communities at refugia sites with an elevation higher than 600 meters. The study by Jackson et. al., concludes that the microbial communities are significantly distinct with the different glaciation histories and putative refugia have distinctive microbial communities when compared with the recently disturbed sites. In my opinion, the study has positive sides and can be accepted for publication, however, there are a few queries and concerns that should be addressed.

1.       Why did the authors exclude archaeal sequences from the analysis, is there any specific reason for this? Although archaea are least abundant in the Antarctic soil, they have important ecological services in this environment such as nitrogen cycling. I wonder if the relative percent of archaeal sequences were up to 3-5% then it will be a significant part excluded from the study to understand the “microbial diversity”.  I suggest including archaea in the study and redoing the analysis.

2.       In the materials and methods, section authors did not specify the number of samples collected from the individual sites (refugia and disturbed). It can be included in table 1.

3.       Is there a reason why the authors performed the NMDS with environmental variables analysis but did not perform the DB-RDA analysis? Did the authors test the community data for heterogeneity? The test of heterogeneity can be performed by DCA analysis in the Vegan package. It can be included in the article.

4.       Line number 427-430: How unweighted unifrac is more informative than weighted unifrac analysis, Is it correct? Authors can also include the weighted unifrac distance-based analysis result in the MS.

Some minor comments:

Line no. 42-43: Which communities, “microbial” or “soil”?

Line no. 92: missing bracket.

Figure 2: Include a figure for the phyla distribution for these sites.

Figure 1: The quality of the figure is not good and the text inside the figure should be improved. Overall all figures need improvement. 

Author Response

Reviewer 3

The manuscript by Jackson et. al entitled “Glacial legacies: Microbial communities of Antarctic refugia” describes the impact of historical glacial and lacustrine disturbance events on microbial communities in the McMurdo dry valleys (MDV). They have reported the effect of historical glacial and lacustrine disturbance events on microbial communities. They also identified similar microbial communities at refugia sites with an elevation higher than 600 meters. The study by Jackson et. al., concludes that the microbial communities are significantly distinct with the different glaciation histories and putative refugia have distinctive microbial communities when compared with the recently disturbed sites. In my opinion, the study has positive sides and can be accepted for publication, however, there are a few queries and concerns that should be addressed.

  1. Why did the authors exclude archaeal sequences from the analysis, is there any specific reason for this? Although archaea are least abundant in the Antarctic soil, they have important ecological services in this environment such as nitrogen cycling. I wonder if the relative percent of archaeal sequences were up to 3-5% then it will be a significant part excluded from the study to understand the “microbial diversity”.  I suggest including archaea in the study and redoing the analysis.

This was a great suggestion – we excluded archaea because of experience with other systems, but we agree that they do merit attention in this environment. Archaeal reads were quite variable across the sites and are now included in the new comparisons. Most figures (2-9 and some of the supplementary figures) were updated, although the resulting community compositions were mostly similar and the results in the manuscript have remained mostly unchanged. We have updated the methods section to reflect these changes and added a few lines in the results sections to include Archaea (L365-366, L572, L784).

The one area where the results varied after including Archaeal reads was in the model selection approach, where a separate analysis of the candidate refugia and recently disturbed sites previously revealed the same results as when the two groups were analyzed together; now the two groups give different results, and this is adjusted in the text at L402-419. The difference is likely driven by a difference in the Shannon diversity metric at the sites in Miers and Garwood valleys and the McKay Glacier basin caused by significant presence of Archaea at these sites.

  1. In the materials and methods, section authors did not specify the number of samples collected from the individual sites (refugia and disturbed). It can be included in table 1.

We’ve updated the materials and methods (L185) to reflect that we took a single sample from each considered site. All samples were analyzed in 6 replicates as described in the L191.

  1. Is there a reason why the authors performed the NMDS with environmental variables analysis but did not perform the DB-RDA analysis? Did the authors test the community data for heterogeneity? The test of heterogeneity can be performed by DCA analysis in the Vegan package. It can be included in the article.

We thank the reviewer for these suggestions – RDA and DCA analyses are now included in the supplementary materials. RDA results are highly congruent with the NMDS ordination plots – but the DCA adds an interesting comparison in terms of sample homogeneity. We have added information on this in the methods paragraph L 289-297, and results L450-454 to reflect this.

  1. Line number 427-430: How unweighted unifrac is more informative than weighted unifrac analysis, Is it correct? Authors can also include the weighted unifrac distance-based analysis result in the MS.

Weighted UniFrac results are included in Figure 4 and PERMANOVAs are reported in the supplementary material. Differences between Weighted UniFrac ordination plots and Weighted UniFrac were minimal, and any differences are reported in L362-366

Some minor comments:

Line no. 42-43: Which communities, “microbial” or “soil”?  

We have added “soil” before communities

Line no. 92: missing bracket.

corrected

Figure 2: Include a figure for the phyla distribution for these sites.

We have included a better-quality figure that makes the distribution of the different phyla at each site clearer.

Figure 1: The quality of the figure is not good and the text inside the figure should be improved. Overall all figures need improvement. 

We will provide a pdf version of the photo for Biology to embed in the document as a higher resolution image